# 50 Gb/s Electro-Absorption Modulator Integrated with a Distributed Feedback Laser for Passive Optical Network Systems



**Daibing Zhou** [1,2,3], **Song Liang** [1,2,3], **Ruikang Zhang** [1,2,3], **Qiulu Yang** [1,2,3], **Xuyuan Zhu** [1,2,3], **Dan Lu** [1,2,3], **Lingjuan Zhao** [1,2,3,*] **and Wei Wang** [1,2,3]

1   Key Laboratory of Semiconductor Materials Science, Institute of Semiconductors, Chinese Academy of Sciences, Beijing 100083, China
2   College of Materials Science and Opto-Electronic Technology, University of Chinese Academy of Sciences, Beijing 100049, China
3   Beijing Key Laboratory of Low Dimensional Semiconductor Materials and Devices, Beijing 100083, China
*   Correspondence: ljzhao@semi.ac.cn; Tel.: +86-10-82304437

**Abstract:** We report an electro-absorption modulator integrated with a distributed feedback Bragg laser fabricated by butt-joint technology. The lasing wavelength of the EML laser was 1311.71 nm, the output power was 10.04 mW when the current of the DFB section was 100 mA, the side-mode suppression ratio was greater than 50 dB, and the small-signal bandwidth was 29.40 GHz when the bias voltage of the modulator was −2 V. A 50 Gb/s data transmission over a single-mode fiber of up to 10 km was realized, which could be used as a light source for 50 G passive optical network systems.

**Keywords:** electro-absorption modulator; multiple quantum wells; data transmission

## 1. Introduction

With the development of high-capacity services such as cloud computing, big data and data centers, the demand for bandwidth of optical communication networks is increasing [1–3]. In February 2018, the International Telecommunication Union-Telecommunication standardization Sector (ITU-T) meeting passed the 50 G time division multiplexing passive optical network standard research and project proposal, and it is expected that 50 G passive optical networks (PON) will be commercially available in 2023. 50 Gb/s electro-absorption modulators integrated with distributed feedback Bragg lasers (EML) will be used as the light source devices for 50 G time division multiplexing passive optical networks [4–8].

Technologies including selective area growth technology [9,10], identical active layer technology [11,12], and quantum well intermixing technology [13,14] have been used for the fabrication of EMLs. Though characteristics, such as a large EAM bandwidth and high extinction ratio have been demonstrated, the EAM multi-quantum wells (MQWs) and laser MQWs cannot be optimized at the same time for these techniques. In comparison, in the butt-joint technology, the growth of laser and modulator MQW materials are in two separate material growth steps. Thus, optimum parameters can be adopted for both kinds of MQWs. As a result, most high-performance EMLs reported recently are fabricated by the butt-joint process [15,16].

However, to obtain high-quality butt-joint interfaces between the laser and EAM, careful butt-joint growth optimization is needed, especially when InGaAlAs is used. InGaAlAs MQWs are easily oxidized during the growth process, forming defects and affecting the performance of the device. During the fabrication of EML lasers, among the different types of fabrication errors, the angle and quality of butt-joint materials have a large impact on the performance of the EML laser. If the material butt-joint is poorly controlled, holes will be generated at the butt-joint interface which directly affects the coupling efficiency and

the reliable performance of the chip, so the corrosion conditions of InGaAlAs materials are strictly controlled.

In this paper, a low-cost ridge waveguide structure was used to fabricate a EML device, and a special process was used to solve the oxidation and defect problems of the material butt-joint process. 50 Gb/s non-return-to-zero (NRZ) data was modulated onto the device and transmitted over a 10 km single mode fiber, which is of great significance for the 50 G PON system.

## 2. Device Fabrication

To fabricate the EML device, the MQW material of the DFB laser was grown on the InP substrate first using metal–organic chemical vapor deposition (MOCVD) equipment. The MQWs consisted of 5 compressively strained InGaAlAs wells and six tensilely strained InGaAlAs barriers. The thickness of the wells and barriers were 5 nm and 9 nm, respectively, sandwiched between two 100-nm-thick separate confinement heterostructure InGaAlAs layers. The photoluminescence peak wavelength was 1300 nm. A 250-nm-thick $SiO_2$ layer was deposited on the MQW layer by plasma-enhanced chemical vapor deposition. $SiO_2$ strip masks were formed in the DFB area along the 110 direction using photolithography and the reactive ion etching process. The strip width of the silicon dioxide was 20 μm. The material outside the mask strip was etched to a thickness of 200 nm using inductively coupled plasma equipment, and then an acid etching solution ($H_2SiO_4$:$H_2O_2$:$H_2O$ = 3:1:1) was used to selectively etch away the remaining InGaAlAs material. Then, HF solution was used for deoxidation, and $(NH_4)_2S$ solution was used for passivation to reduce the oxidation and defects of butt-joint interface materials. Then, the MQW material for the modulator was grown by a second MOCVD process. The MQW material of the modulator consisted of five 10-nm-thick well layers and six 5-nm-thick barrier layers, and the other material structures were the same as the laser. The thickness of the wells and barriers were 10 nm and 5 nm, respectively. The photoluminescence peak wavelength of the EAM MQWs was 1260 nm. Gratings were formed in an InGaAsP layer over the MQW material of the DFB region using electron beam exposure equipment. Finally, a P-type InP cladding layer and an InGaAs contact layer were grown.

A reverse-mesa ridge-waveguide structure was formed using the photolithographic and wet etching technology, having a 3 μm ridge width. The lengths of the laser and modulator sections were 450 μm and 100 μm, respectively. The two sections were electrically isolated by a 50 μm long region, in which $He^+$ was implanted. The parasitic capacity of the modulator was reduced by forming a polyimide layer under the contact pad. Ti–Au thin films were deposited as a p-electrode using the lift-off technique. A Au–Ge–Ni n-electrode was deposited as an n contact metal. A rapid thermal annealing process was applied to form an ohmic contact. A schematic view of an EML laser is shown in Figure 1. The laser was bound to an AlN submount for testing, during which, the temperature of the device was maintained at 25 °C by a thermoelectric cooler. An optical image of an EML laser is shown in Figure 2.

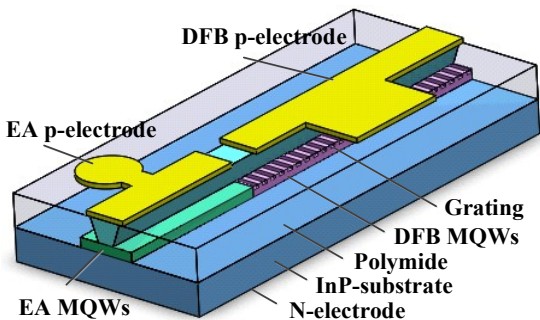

**Figure 1.** Schematic view of an EML laser using butt-joint technology.

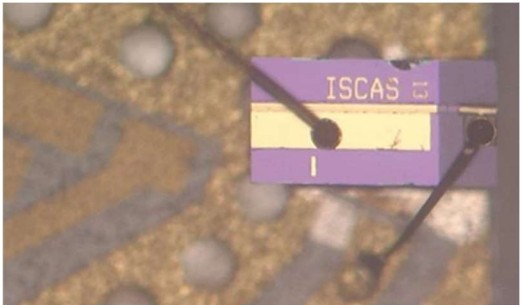

**Figure 2.** An optical image of an EML laser bound to an AlN submount.

### 3. Experimental Setup and Results

Figure 3 shows the light–current–voltage characteristics of the EML laser at 25 °C when the EAM modulator was biased at 0 V. The threshold current of the laser was 15 mA, and the output power was 10.04 mW at 100 mA. When a current of 70 mA was applied to the DFB section and the modulator was biased at 0 V, the lasing wavelength of the EML laser was 1311.71 nm, with a greater than 50 dB side mode suppression ratio (SMSR), as shown in Figure 4.

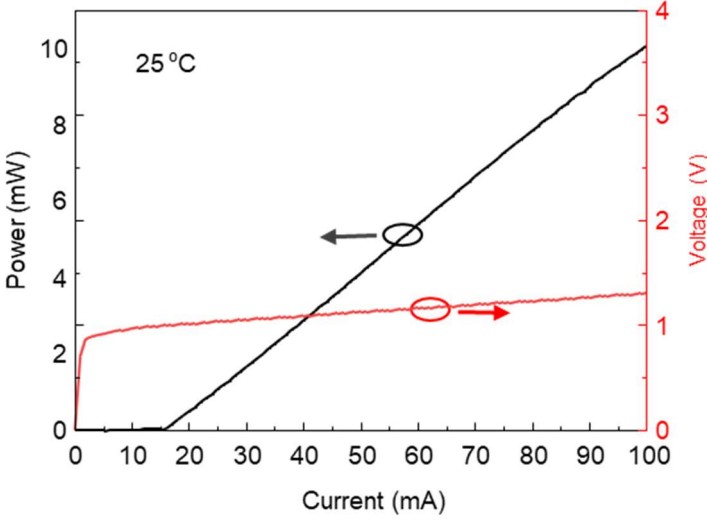

**Figure 3.** The light–current–voltage characteristics of the EML laser at 25 °C.

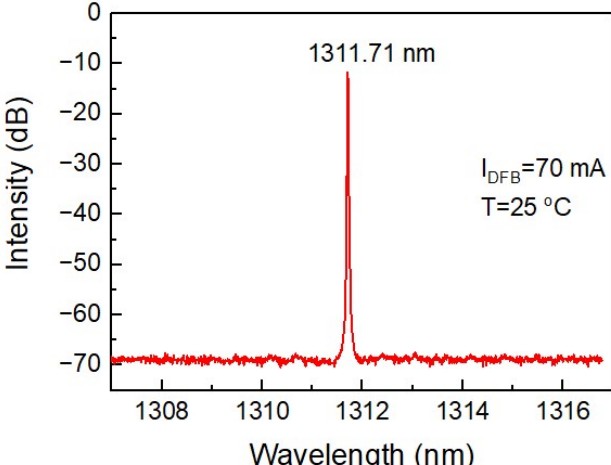

**Figure 4.** Typical optical spectrum of the DFB section of the EML laser at 70 mA.

Figure 5 shows the static extinction ratio characteristics of the EML laser. During the test, the current of the DFB section was fixed at 70 mA and the reverse voltage applied to

modulator was varied. The light of the EML laser was coupled to an optical power meter through a single-mode fiber. From Figure 5, it can be seen that the static extinction ratio of the EML laser is greater than 25 dB under a reverse bias voltage of −5 V.

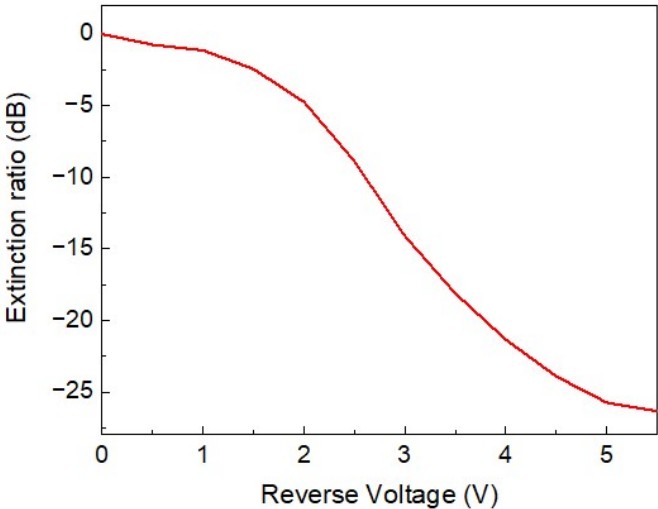

**Figure 5.** Extinction ratio of the EML laser versus the modulation bias.

The small-signal frequency response of the EML laser was measured using a HP 8510C vector network analyzer. A Finisar U2T-XPDV2320R photodetector was used for optical-to-electrical signal conversion. In the experiment, a current of 70 mA was applied to the DFB section and the bias voltage of the modulator was −2 V. A 50 ohm resistor was connected in parallel with the modulator for impedance matching. The light output of the EML laser was coupled into the detector through a single-mode fiber. The modulation response of the EML laser is shown in Figure 6. The 3 dB bandwidth of the modulator was 29.4 GHz.

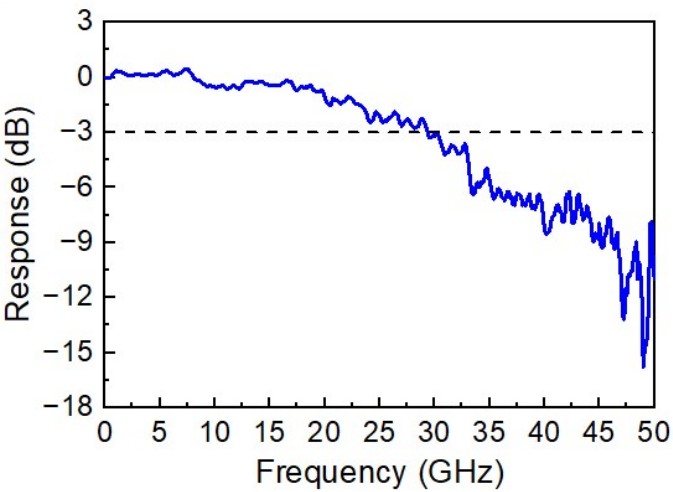

**Figure 6.** Small-signal modulation response of the EML laser when the DFB current was 70 mA and the bias voltage of the modulator was −2 V.

Data transmissions using the device were conducted using standard single mode fibers. A current of 70 mA was applied to the DFB section and the modulator was biased at a voltage of −2 V. The modulation voltage swing (Vpp) was 2 V. The modulated light power for the 25 and 50 Gb/s modulation were 2 mW. The type of sampling scope used was the Keysight 86100D. Figure 7a shows the back-to-back (BTB) eye diagrams under 25 Gb/s NRZ pseudorandom binary sequence (PRBS $2^{15}$-1) signal modulation. Clearly opened eyes were obtained and had a 5.3 dB dynamic extinction ratio. Figure 7b shows

the eye diagrams after 25 km fiber transmission at 25 Gb/s modulation. The eye diagrams somewhat deteriorated but remained clear. Figure 7c,d shows the eye diagrams for BTB and 10 km transmissions, respectively, when the device was modulated by 50 Gb/s NRZ data. For the BTB condition, the eyes are clear and the dynamic extinction ratio was 4.6 dB. However, the eye quality deteriorates significantly after 10 km transmissions. At a higher data rate, the chromatic effects of the fiber were more prominent, leading to a worse transmission performance.

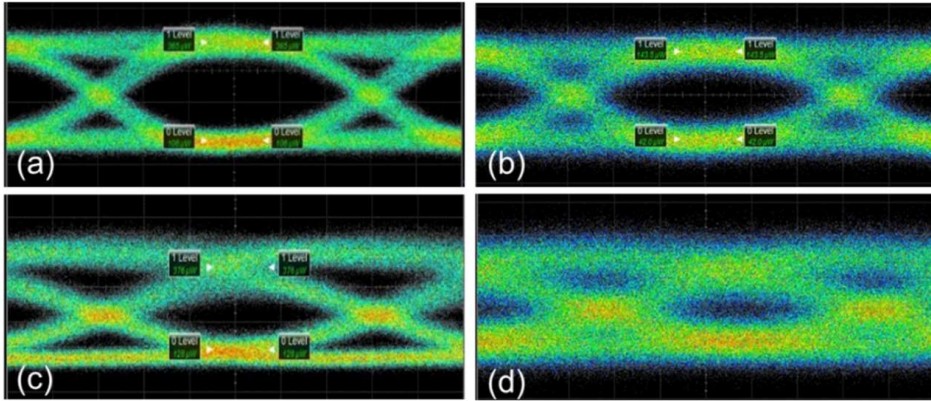

**Figure 7.** Eye diagrams of 25 Gb/s and 50 Gb/s: (**a**) 25 Gb/s BTB eye diagram, (**b**) 25 Gb/s eye diagram after 25 km transmission, (**c**) 50 Gb/s BTB eye diagram, (**d**) 50 Gb/s eye diagram after 10 km transmission.

## 4. Discussion

With the development of PONs, EML lasers in the 1.3 micron wavelength band will be widely used. Unlike the InGaAsP material, the InGaAlAs quantum wells have more favorable conduction band offset and is usually used in EML lasers in the 1.3 μm band. However, the InGaAlAs material is prone to oxidation and defects during fabrication. In this work, a special treatment process was used in the material butt-joint process and a ridge waveguide structure was used, which exposed less InGaAlAs material to air compared to buried structures [15]. Compared with previous works [17], in which two butt-joint processes were used for the fabrication of EMLs, only one butt-joint process was needed for our device, which helps to lower the fabrication costs. With the butt-joint technique the laser and modulator can be optimized separately, and the absorption efficiency of the 100-μm-long modulator is comparable to that of the 150-μm-long modulator of the selected growth area [9]. However, this comes with an increase in the small-signal bandwidth of nearly 10 GHz. The fabricated EML chips have potential applications in the future development of 50 G PON systems.

## 5. Conclusions

In summary, an electro-absorption modulator integrated with a distributed feedback laser was fabricated by butt-joint growth technology. The small-signal modulation bandwidth of the modulator was 29.4 GHz. Clear open eye diagrams were obtained under 50 Gb/s NRZ signal modulation with a dynamic extinction ratio greater than 4.6 dB in BTB conditions. The EAM laser is a promising light source for the next-generation 50 G time division multiplexing passive optical network systems.

**Author Contributions:** Conceptualization, S.L. and L.Z.; methodology, D.Z.; experimental work and data analysis, D.Z. and R.Z.; measurement, Q.Y., X.Z. and D.Z.; writing—original draft preparation, D.Z.; writing—review and editing, D.L.; visualization, D.L.; supervision, W.W.; funding acquisition, D.Z. All authors have read and agreed to the published version of the manuscript.

**Funding:** The research is funded by the National Key Research and Development Program of China 2019YFB1803801, Beijing Municipal Natural Science Foundation 4212056, National Natural Science Foundation of China 61974165 and 62274156.

**Institutional Review Board Statement:** Not applicable.

**Informed Consent Statement:** Not applicable.

**Data Availability Statement:** The data presented in this paper are available on the request form the corresponding author.

**Conflicts of Interest:** The authors declare no conflict of interest.

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
