# Peer review of "50 Gb/s Electro-Absorption Modulator Integrated with a Distributed Feedback Laser for Passive Optical Network Systems"

_photonics, doi:10.3390/photonics9100780_

Round 1
Reviewer 1 Report
Please see the attachment

Reviewer 2 Report
I attched comments.
TRANSLATE with x English| Arabic | Hebrew | Polish |
| Bulgarian | Hindi | Portuguese |
| Catalan | Hmong Daw | Romanian |
| Chinese Simplified | Hungarian | Russian |
| Chinese Traditional | Indonesian | Slovak |
| Czech | Italian | Slovenian |
| Danish | Japanese | Spanish |
| Dutch | Klingon | Swedish |
| English | Korean | Thai |
| Estonian | Latvian | Turkish |
| Finnish | Lithuanian | Ukrainian |
| French | Malay | Urdu |
| German | Maltese | Vietnamese |
| Greek | Norwegian | Welsh |
| Haitian Creole | Persian |

Reviewer 3 Report
1. It seems that this manuscript is a Letter. Many experimental details can be provided in order to facilitate readers.
2. It seems that the eye diagrams of Fig. 7 are obtained by Keysight sampling scope. The model of the scope should be given because each model includes filters which can generate important influence on the eye diagram.
3. Fig. 4, the span of the optical spectrum is about 8 nm. Is there any sidelobe for a bigger span?
4. Line 93, HP 8510C is a very old equipment. The input and the output signals are electrical. That is to say, the author should use a photodiode placing at the input of the network analyzer to obtain the results in Fig. 6.
5. Line 113, the comma after modulated by 50 Gb/s NRZ data should be period.
Reviewer 4 Report
Overall, the manuscript is well written and may be acceptable after major revision.
1. The title and abstract shouldn't contain any abbreviations. Give the abbreviation of "NRZ" in the introduction. Full form of all abbreviations should be written when it comes the first time into the manuscript. Besides, the manuscript contains too many abbreviations, please reduce the number of abbreviations.
2. The last paragraph of the introduction should cover the scientific details and the novelty of the work.
3. Although there are many fresh publications in this field, you used only one reference in 2022.
4. Proper justification is not given for the characteristics of the results.
Round 2
Reviewer 2 Report
All comments and suggests were answered.
TRANSLATE with x English| Arabic | Hebrew | Polish |
| Bulgarian | Hindi | Portuguese |
| Catalan | Hmong Daw | Romanian |
| Chinese Simplified | Hungarian | Russian |
| Chinese Traditional | Indonesian | Slovak |
| Czech | Italian | Slovenian |
| Danish | Japanese | Spanish |
| Dutch | Klingon | Swedish |
| English | Korean | Thai |
| Estonian | Latvian | Turkish |
| Finnish | Lithuanian | Ukrainian |
| French | Malay | Urdu |
| German | Maltese | Vietnamese |
| Greek | Norwegian | Welsh |
| Haitian Creole | Persian |
Reviewer 3 Report
accept
Reviewer 4 Report
Accept in present form